# Viral Hepatitis in Pregnant Mexican Women: Its Impact in Mother–Child Binomial Health and the Strategies for Its Eradication

**DOI:** 10.3390/pathogens13080651

**Published:** 2024-08-02

**Authors:** Carmen Selene García-Romero, Carolina Guzmán, Alejandra Martínez-Ibarra, Alicia Cervantes, Marco Cerbón

**Affiliations:** 1Departamento de Infectología e Inmunología, Instituto Nacional de Perinatología Isidro Espinosa de los Reyes, Mexico City 11000, Mexico; selun_gem@yahoo.com.mx; 2Laboratorio de Hígado, Páncreas y Motilidad, Unidad de Medicina Experimental, Hospital General de México Dr. Eduardo Liceaga, Mexico City 06720, Mexico; carolina.guzman@quimica.unam.mx; 3Facultad de Química, Universidad Nacional Autónoma de México, Mexico City 04960, Mexico; alejandra_martinez@hotmail.es; 4Servicio de Genética, Hospital General de México Dr. Eduardo Liceaga, Mexico City 06720, Mexico; acervant@unam.mx; 5Facultad de Medicina, Universidad Nacional Autónoma de México, Mexico City 04960, Mexico

**Keywords:** viral hepatitis, pregnancy, eradication, prevention, vertical transmission, pregnant Mexican women

## Abstract

Viral hepatitis is the main cause of infectious liver disease. During pregnancy, a risk of vertical transmission exists both during gestation and at birth. HAV, HBV, and HCV might progress similarly in pregnant and non-pregnant women. In this study, we found a prevalence of 0.22% of viral hepatitis in pregnant women, with a light preponderance of HCV over HAV and HBV. Here, it was observed that acute HAV infection is more symptomatic and has higher risks for the mother and fetus, in a similar manner to what has been reported for HEV. Histopathological alterations were observed in all except one placenta, indicating that it is an important tissue barrier. Regarding the Mexican strategies for viral hepatitis eradication, success may be related to vaccination at birth, whereas for HCV, the national program for eradication is aimed at treating the infection via direct-acting antiviral agents. The HBV strategy has positively impacted pregnant women and their children, diminishing the risk of vertical transmission. The HCV strategy is still in its early years, and it is expected to be just as successful. For acute hepatitis, HAV and HEV, programs promoting hand washing and those aimed at providing clean food and water are applicable as preventive strategies, alongside other programs such as vaccination.

## 1. Introduction

Viral hepatitis is the main cause of infectious liver disease; five hepatotropic viruses are recognized as being responsible: the hepatitis A (HAV), B (HBV), C (HCV), D (HDV), and E (HEV) viruses [1]. The course of the infection depends on the virus affecting the liver; in immunocompetent patients, HAV and HEV are associated with acute hepatitis and short-term disease [2]. In contrast, HBV and HCV are bloodborne infections and produce chronic disease and consequent liver damage, including inflammation, fibrosis, cirrhosis, liver failure, and hepatocellular carcinoma (HCC), whereas HDV is a defective virus, only able to produce infection when HBV is present, increasing the risk of HCC [2,3,4].

The prevalence around the world is heterogeneous for each type. HAV and HEV show increased prevalence in developing countries, mainly associated with fecal–oral transmission and water and food contamination [2]. HBV and HDV are highly prevalent in the Western Pacific and Africa, and HCV is globally spread, but its genotypes are geographically distributed [2]. Regarding the five viruses producing viral hepatitis, A and B, and, therefore, D, are preventable by vaccination. A vaccine has been developed for HEV in China; however, it has not been approved in other countries. There is no available vaccination for HCV [2,3].

During 2020, the Mexican government reported that HAV showed an incidence rate of 3.11 cases per 100,000 inhabitants. HBV infection was very low due to universal vaccination programs, showing an incidence of 0.28 cases per 100,000 inhabitants, whereas HCV showed an incidence of 1.06 cases per 100,000 inhabitants [5]. A variable prevalence of HEV has been reported in different regions of Mexico with a dependence on the age group, ranging from 3.4 to >15% [6]. Regarding HCV, recent changes in public health policies have been approved that are intended to provide universal treatment for HCV infection through direct-acting antiviral agent (DAA) therapies. With this approach, it is expected that HCV will be eradicated; however, it has been noted that even when the viral infection is cured, the liver damage is able to progress to HCC in patients with severe fibrosis and cirrhosis [7].

Pregnancy is considered a physiologic state involving enormous changes, including endocrine and immunologic adaptations aimed to assure the development and growth of the fetus. Due to these changes, during pregnancy, viral hepatitis can be life-threatening to both mother and child, depending on the stage of gestation and the virus affecting the mother [1,8]. Although all hepatitis viruses can harm the mother and the child, acute viral hepatitis is the most frequently documented cause of jaundice in pregnant women [9] and the greatest risk to maternal health, and the fetus is subsequently seen with acute HAV or HEV infection. Infection with HEV genotype 1/2, especially in the third trimester, is associated with fulminant hepatitis and a mortality rate of 30% [1,10,11]. By contrast, the primary risks for HBV, HCV, and HDV are related to the severity of the underlying liver disease in the mother and the risk of mother-to-child transmission (MTCT) for HBV and HCV. The prevention of MTCT is key to reducing the global burden of chronic viral hepatitis, and prevention strategies must take into consideration local healthcare and socioeconomic challenges [1].

There are few reports of the prevalence of viral hepatitis during pregnancy in the Latin American population, particularly that in Mexico [12]. The aim of this report is to review the prevalence of viral hepatitis in pregnant Mexican women, its potential impact on the mothers and their children, and the strategies for the elimination of viral hepatitis in Mexico.

## 2. Patients and Methods

### 2.1. Patients

This was a descriptive and retrospective study; we included pregnant Mexican women with a serological diagnosis of viral hepatitis who were attended between January 2018 and December 2023 at the Instituto Nacional de Perinatologia Isidro Espinosa de los Reyes (INPer) in Mexico City. This is a specialized tertiary referral hospital for high-risk pregnancies. The inclusion criteria were (1) pregnancy and (2) a serological diagnosis of viral hepatitis: HBsAg positive for HBV or the presence of antibodies against HCV or HAV. The exclusion criterion was a diagnosis of autoimmune hepatitis.

This study did not require Ethics Committee approval because patient data were collected from electronic medical records with anonymization of the individuals. In addition, all patients signed an authorization for the use of their medical data at the time of admission to the hospital. The procedures were in accordance with the ethical standards of the Declaration of Helsinki (1964, amended most recently in 2013) [13].

### 2.2. Variables Extracted from Medical Records

The extracted demographic data included the maternal age, pregnancy history, gestational age at hepatitis diagnosis, hepatic symptomatology, laboratory function tests, and comorbidities (such as obesity, type 1 or 2 diabetes mellitus, HIV co-infection, systemic hypertension, or a history of injected drug use). The data from laboratory tests included HAV, HBV, and HCV serological markers; alanine aminotransferase (ALT); aspartate aminotransferase (AST); total bilirubin (TBIL); direct bilirubin; albumin (ALB); prothrombin activity (PTA); the prothrombin time (PT); and hemoglobin and platelet counts. Pregnancy complications for the mother, such as gestational diabetes, preeclampsia, thrombocytopenia, and anemia, were recorded. Maternal and neonatal outcomes were recorded from the delivery records. The extracted information included the delivery mode (eutocia/cesarean), oligohydramnios, postpartum hemorrhage, premature rupture of membranes, spontaneous abortions, premature delivery, low-birth-weight infants, fetal distress, neonatal asphyxia, newborns admitted to neonatal intensive care unit (NICU), and stillbirths. We also obtained the histomorphological data of the placentas.

### 2.3. Statistical Analysis

We divided our cohort into three groups according to the type of viral hepatitis: A, B, or C. Variables were expressed as the mean ± SD or median (IQR) for numerical variables and as the frequency and percentage for categorical variables.

## 3. Results

Serological screenings for HBV surface antigen (HBsAg) and for antibodies to HCV were performed for all pregnant women attended at INPer in their first obstetric consultation, according to Mexican guidelines [14,15,16]. An HAV diagnosis was made in pregnant women with the symptomatology of acute viral hepatitis and was performed at INPer by the detection of IgG or IgM against HAV. The gestational age at diagnosis for this group (25 ± 8.03 weeks of gestation) was more advanced compared with the age of diagnosis of HBV and HCV, which is almost always done by screening in asymptomatic women, even before pregnancy (Table 1). Additionally, hepatic function tests were performed in the first obstetric consultation.

We identified 38 pregnant Mexican women with a diagnosis of viral hepatitis from a total of 17,206 pregnancies in the study period, for a prevalence of 0.22%. The most frequent viral hepatitis type was HCV with 16 positive women, representing 42.1% of the cohort and a prevalence of 0.093%. For HAV and HBV, there were 11 pregnant women for each (28.9%) and a prevalence of 0.064%. Table 1, Table 2 and Table 3 show the general characteristics of our cohort, the most frequent comorbidities and complications in pregnant women, and the pregnancy and neonatal outcomes in Mexican women with viral hepatitis A, B, or C. The placental histopathological data are shown in Table 4. It was not possible to obtain the complete data for all patients. The total number of pregnant women or children with each characteristic is indicated in the tables and was considered for the statistical analyses.

This retrospective study had serious limitations as it was not possible to obtain all the variables from all the pregnant women included in the cohort, particularly those related to the serological tests realized for the viral hepatitis diagnosis. There were no data available on the viral loads or the virus genotypes.

In the discussion section, we present each type of studied hepatitis separately, including its results and interpretation, to make it easier for the reader. 

## 4. Discussion

### 4.1. HAV in Pregnancy

There are scarce data regarding HAV’s prevalence and incidence during pregnancy in México; we previously reported only two cases (0.05%) of HAV infection during pregnancy in a three-year study that included 10,762 pregnancies at INPer [9]. The prevalence found in the present report is similar, 0.064%, and both are lower than the frequency in other populations [17] but greater than the incidence in the general Mexican population (3.11/100,000) [5]. It has been suggested that HAV infection in the second or third trimester is related to preterm labor, including premature contractions, placental abruption, premature rupture of membranes, and vaginal hemorrhage [18,19]. In our cohort, there were 4/11 preterm births, 2 with premature rupture of membranes and 3 with postpartum hemorrhage, and half of the babies were small for their gestational age.

Intrauterine transmission has not been documented, and vertical transmission during delivery is rare, although possible [1,19], and generally associated with complications during childbirth including meconium, fetal ascites, and neonatal jaundice. Generally, full resolution of the infection occurs with no further risk of death for both mother and newborn [19]. In our cohort, 88.9% of the children presented jaundice with elevated bilirubin, and one of the preterm low-birth-weight infants was in the neonatal intensive care unit and died at 10 days old. Treatment of HAV during pregnancy should not be any different to the standard treatment to avoid maternal and child complications.

Regarding lactation, it has been noted that anti-HAV antibodies are present in breast milk. HAV RNA is also found in maternal milk; nevertheless, there is no evidence suggesting possible transmission of HAV through breastfeeding [20,21]. In fact, maternal lactation might induce protection against the infection in the newborn; however, some results indicate poor vaccine efficiency, particularly in inoculations with inactivated virus presentations, which may imply the need for immunoglobulin treatment against HAV and vaccination after the first year of age [21].

### 4.2. HBV and HDV in Pregnancy

HBV proceeds as a chronic infection, responsible for induced fibrosis, cirrhosis, and hepatocellular carcinoma [22,23]. Coinfection with HDV increases the risk of cancer; however, HDV is unable to infect by itself. In fact, HDV uses HBV molecules to assemble its envelope and succeed in infection [23]. The actual prevalence of HBV infection in pregnant Mexican women is very low. Compared with the rate reported in 2003 of 1.65% [12], we reported a total of five HBV cases (0.05%) in our previous three-year study [9], similar to the 0.064% found in the six-year cohort. The identification of HBV infection during pregnancy is crucial due to its high risk of vertical transmission [24], which occurs both through the placenta and at delivery, causing fetal or neonatal hepatitis [24,25]. This early exposure increases the risk of cirrhosis and HCC at young ages. Evidence shows that an infection acquired at the end of pregnancy increases the probability of developing chronic disease; perinatal HBV infection significantly increases the risk of chronic HBV, in contrast to exposure later in life, and increases the risk of transmission to offspring [17,26]. However, HBV infection during pregnancy, either acute or chronic, does not increase other risks, including those of abortion, stillbirth, or congenital malformation, but a low birth weight might be present [1,17]. We found 4/12 (33.3%) premature babies, and 25% were small for their gestational age; additionally, half presented jaundice and elevated bilirubin. Gestational diabetes was present in 5/11 pregnant women with chronic HBV infection, and only 1 presented postpartum hemorrhage (Table 2 and Table 3).

Maternal lactation is generally recommended for mothers carrying HBV; however, transmission through breast milk is a concerning risk [1,17,27]. Immunoprophylaxis, either with immunoglobulin against HBV surface antigen (HBsAg) or with vaccination, is a plausible option to prevent transmission through breast milk. Zhang et. al., showed that there is an equal risk of transmission in breastfed babies compared with those receiving formula [27], which is in support of breastfeeding. Recently, tenofovir, the first-line drug against HBV infection, was proven effective in preventing MTCT when used as tenofovir alafenamide fumarate or tenofovir disoproxil fumarate; however, tenofovir alafenamide fumarate showed higher renal and breastfeeding safety [28]. Tenofovir has been documented in breast milk at lower levels compared to those observed in cord blood and amniotic fluid, with no differences in growth observed in infants exposed in utero [29].

### 4.3. HCV in Pregnancy

HCV infection used to be among the most common causes of liver disease, producing acute liver injury followed by chronic infection, steatosis, inflammation, fibrosis, cirrhosis, and HCC [22,30]. Since the development of DAAs, this infection is curable by a pharmacologic treatment consisting of 8–16 weeks of therapeutic combinations of an inhibitor of protease NS3/4A, an inhibitor of the non-structural protein NS5A, or an inhibitor of the polymerase NS5B [31]. Before DAA treatment, Mexico reported an estimated seroprevalence of 1% for HCV, mostly composed of genotype 1 subtype b [32]. Among pregnant women, HCV is an important concern in countries with high rates of HIV and injected drug abuse [33,34,35]. The prevalence of HCV infection among the general population in Mexico City is about 1.2–1.4%, based on a systematic review [36]. This rate is higher than that in pregnant women; for example, our previous study at INPer, CDMX, reported 13 cases (0.12%) in a three-year study [9]. However, in the present study, the rate was 0.09%, suggesting a lower prevalence than in the general population.

Maternal infection with HCV possesses a risk of MTCT, occurring in approximately 3–8% of deliveries from mothers with HCV viremia [33,35]; in women coinfected with HCV and HIV, the risk significantly increases to close to 20% [33]. Adverse perinatal outcomes might be present in HCV-infected mothers, including an increased risk of premature delivery [10]; however, few data exist, and confounding factors prevent us from reaching adequate conclusions [35]. Importantly, in our cohort more than 50% of the mothers from the HCV group were of advanced maternal age; this may be the reason for the high proportion of them being asymptomatic, and five were diagnosed before pregnancy. Remarkably, the only miscarriage occurred in this group.

On the other hand, the most frequent maternal complication in this group was thrombocytopenia. Regarding the babies, four of them were preterm, one required NICU admission, and 67% presented jaundice.

According to the Mexican guidelines, infants from mothers with high viral loads at delivery should be tested before 18 months of age for antibodies against HCV. A positive test should be confirmed before 3 years of age by a viral RNA test. HCV-positive children weighing >30 Kg are candidates for DAA treatment adjusted to their liver damage, age, and body weight [16].

Lactation is not contraindicated for these mothers; however, care should be taken if patients present skin, areola, and nipple lesions or bleeding. In those cases, lactation should be avoided until the lesions are completely recovered. Lactation is not recommended in the case of coinfection with HIV [14,16].

### 4.4. HEV in Pregnancy

HEV is distributed globally, and eight genotypes (1–8) have been identified. Genotypes 1–4 infect humans and can be transmitted from person to person, whereas genotypes 5–8 are associated with zoonotic infection via the consumption of undercooked meat and liver from infected animals, including pig, wild boar, camel, fox, chicken, rat, and rabbit, among others [37,38]. Outbreaks have been reported, and the main cause of transmission is fecal–oral by the ingestion of contaminated food and water [37,39]; although HEV is generally self-limited and asymptomatic, it can also proceed as acute hepatitis or produce chronic disease, showing cirrhosis in a short period in organ recipients in association with other immunosuppressive conditions [37,39,40].

In contrast, pregnancy, particularly the third trimester, represents a high risk of developing fulminant hepatitis with an elevated rate of mortality [11,17,41]. Preterm labor, abortion, a high risk of mortality for both the mother and fetus, and even a risk of mortality in newborns are among the complications associated with HEV infection during pregnancy; the maternal mortality rate is estimated at 20% [17,41,42]. MTCT is also observed. HEV RNA has been reported in the cord and peripheral blood of newborns from infected mothers [11], and it has been suggested that the transmission rate might reach 33–100% [1]. HEV antibodies and viral RNA presence have been reported in colostrum; however, breastfeeding seems to be safe [1,17].

Given the high incidence of HEV worldwide [41] and the high MTCT rate of over 40% [17], HEV infection therefore implies a high risk of fulminant hepatitis during pregnancy, with the clinical and biochemical features of fulminant hepatitis. It is well known that HEV is the most frequent acute viral hepatitis around the world. In Mexico, HEV genotypes 1 and 3 are the most frequent, while HEV G2 is currently not found [6]. The prevalence of HEV antibodies in pregnant Mexican women was determined in two independent studies conducted in Durango and Mexico City, showing prevalence rates of 5.7% and 7.4%, respectively [38,43].

The underestimated cases of viral hepatitis in pregnant Mexican women reported here (0.22%) should be related to the fact that, unfortunately, in Mexico, there is no regular screening for HEV infection. Thus, it is mandatory that pregnant Mexican women with acute hepatitis also be screened for this viral type, and it must also be determined in women positive for chronic hepatitis HBV or HCV, as the presence of HEV could produce adverse outcomes for the mother and child [44].

### 4.5. Placental Alterations in Viral Hepatitis

It is well known that the placenta is a barrier for infectious diseases during gestation, including hepatitis. All hepatitis viruses infect and grow in placental tissue, affecting its functionality, immune response, and structure to different degrees. Indeed, HAV, HBV, and HEV cross and replicate in the placental tissue in 80% of patients with these diseases. In contrast, HCV presents lower values of infected placentas, approximately 5%, but coinfections with another virus such as HIV or HBV increase its infectivity to 10 to 15%. In addition, MTCT has been detected in many of them. HAV passes more frequently to the baby, inducing neonatal hepatitis but without long-term health problems [45].

HBV infection of the fetus is associated with placental infection. HBV infection of amniotic cells is an important factor of intrauterine infection. The placental barrier can protect the fetus from infection to some extent. When this immunological barrier fails, some histopathological changes in the placental tissue induce damage and tissue alterations, for example, fibrinoid necrosis, chorionic hyperemia, and a decreasing number of Hofbauer cells, which may play a role in fetal hepatitis infection [46,47].

Other reports also demonstrated the histopathological changes that occur following HBV infection of the placenta, including stromal fibrosis, syncytial knotting, fibrinoid deposition, fibrinoid necrosis, villous capillary congestion, and proliferation [48].

Another cohort study including 101 full-term placentas from HBsAg-positive pregnant women reported that HBV infection rates decreased in cell layers from the maternal side to the fetal surface. Large deposits of bilirubin can be seen in Hofbauer cells (fetal macrophages), trophoblasts, and macrophages of the fetal and extraplacental membranes in active HBV infections [49]. In our study, placentas from infected patients presented tissue damage, and the data indicated that the placentas were more damaged by HAV, followed by HBV and, finally, HCV; however, all presented both maternal vascular malperfusion and fetal vascular malperfusion, without significant differences between the groups, probably due to the size of the sample. The placental damage was characterized by hypotrophy, infarcts, accelerated villous maturation, and increased syncytial knots, similar to that in the other studies mentioned above. However, the most frequent alteration observed was chorangiosis, and some differential alterations were observed depending on the viral type, such as stem vessel obliteration, infarcts, and retroplacental hemorrhage, without significant differences. More studies are required to determine whether there are placental differences according to the severity of hepatitis disease.

### 4.6. Mexican Strategies to Eliminate Viral Hepatitis and Their Impact on Pregnant Women and Their Children

In Mexico, the most important strategy for HAV is prevention, based on hygiene improvements and hand hygiene recommendations; additional strategies include providing clean food and water and avoiding fecal–oral transmission [16]. Although a vaccination is available, recommended in two doses at a basal time and 6 months later for adults at risk [16], the HAV vaccine is not part of the national scheme applied to Mexican children; however, HAV vaccination might be recommended to pregnant women depending on specific circumstances, despite pregnancy not increasing the risk of developing liver damage or failure. The HAV vaccine contains attenuated virus, but it is safe to be applied during pregnancy. Its administration does not induce disease, and it is indicated for pregnant women who are at risk of exposure to the virus, either prior to or after exposure, and for women who frequently travel abroad to high-prevalence countries. In these cases, immunization is recommended 6 months prior to pregnancy, but it is considered safe during pregnancy [50,51]. In addition, a program has been put in place by the Mexican National Ministry of Health to eradicate HAV, including vaccines and treatments.

Prevention is the most impactful strategy for HBV and, therefore, HDV, and it is based on universal vaccination and maternal immunization. In 1999, Mexico started vaccinations against HBV with a complete-cell pentavalent vaccine, and by 2000, this strategy had become massive, reaching adolescents and adults. Nowadays, a recombinant DNA vaccine against the surface antigen of HBV is administered to all children. The most frequently used scheme consists of three doses: the first is applied from birth to 7 days postpartum, the second dose is applied at 2 months, and the third at 6 months of age. For newborns unable to be vaccinated in the first week of life, it is possible to receive the doses at 2, 4, and 6 months. This strategy has led our country to exhibit a very low prevalence of HBV infection and, therefore, a very low prevalence of HDV coinfection [5]. For pregnant women, the HBV vaccine is highly recommended. HBV might increase in severity during the third trimester, increasing the risk of spontaneous abortion, preterm labor, and neonatal hepatitis [50,51]. HBV vaccination is safe for both the mother and fetus and can be applied from 6 months prior to or during pregnancy; for newborns exposed to HBV, vaccination should be administered as soon as possible.

Another strategy to avoid HBV spreading is blood testing, which is mandatory for blood donors and highly recommended for pregnant women who have not received vaccination against HBV; an acknowledgement of the status of infection is important to prevent MTCT [16]. Other preventive strategies are based on education and increasing awareness of maintaining healthy habits, including sexual protection and not sharing needles and other objects of personal hygiene, among others.

The preventive strategies for HCV are those shared with other infections spread by blood, including the safe blood transfusions, safe sex, avoiding the reuse or inadequate sterilization of medical equipment, and not sharing needles and other objects of personal hygiene [16]. There is no vaccine available for HCV; however, a very successful treatment has been developed in the last decade. In fact, only HCV has a recognized effective pharmacologic treatment with DAAs [31]. In 2020, Mexico joined the WHO’s world strategy to eliminate HCV and launched the National Program for Elimination of Hepatitis C, intended to provide access to tests and treatment; a national registry through a unified health system, focusing on the first level of attention; and focalized strategies for high-risk groups [52]. The scheme of treatment is aimed at reducing the morbi-mortality associated with cirrhosis and HCC, with the benefits of lowering contagion. The therapy consists of Sofosbuvir 400 mg/Velpatasvir 100 mg every 24 h for 12 weeks, or Glecaprevir 400 mg/Pibrentasvir 100 mg for 8 weeks in patients without cirrhosis or with compensated cirrhosis; some other therapeutic schemes are available for patients failing to respond [52].

All pregnant women should be screened for HCV at their first antenatal visit. For women infected with HCV who are intending to become pregnant, it is recommended to take the treatment either before becoming pregnant or immediately postpartum [16]. Vertical transmission should be prevented by avoiding any test or procedure that increases the possibility of contact of maternal and fetal blood, during both pregnancy and delivery. However, children older than 3 years are candidates to receive either Sofosbuvir/Velpatasvir or Glecaprevir/Pibrentasvir in doses and times according to their body weight and age [16]. Recent data regarding DAAs have suggested that they are safe to consume during pregnancy [31]. A phase I study showed that the use of ledipasvir combined with sofosbuvir to treat pregnant women is safe, is well tolerated, prevents MTCT, and provides a 100% sustained antiviral response [34]. This evidence is promising; however, more studies are needed to ascertain other variables, including coinfection with HIV. HCV testing is mandatory for blood donation, as are HBV and HIV testing; other strategies are based on education on the risk factors and how to avoid them [16].

HEV is potentially life-threatening in pregnant women, especially when infection occurs in the third trimester, due to the high likelihood of developing fulminant hepatitis [21,37,39]. As for HAV, the most important strategies to prevent HEV infection are maintaining good hygiene practices such as hand washing, ensuring clean food and water, and avoiding undercooked meat and liver [16]. In Mexico, there is no available vaccination for this disease. HEV infection has been overlooked; it is important to increase awareness regarding the importance of HEV prevention, especially among pregnant women, and to improve education aimed to ensure hygienic conditions and sanitary facilities all over the country, with a special emphasis on rural locations. Accordingly, it is important to consider testing for HEV infection in pregnancies complicated with acute hepatitis in the third trimester with a risk of fulminant liver failure.

## 5. Concluding Remarks

In this report, we found a prevalence of viral hepatitis of 0.22% among pregnant women, with a light preponderance of HCV over HAV and HBV. However, there were only two fatal outcomes: a miscarriage in a pregnant woman with HCV infection and the death of a child in the HAV group. Placental histopathological alterations were observed in all except one of the studied patients, indicating that the placenta is an important infected and replicative viral tissue barrier. It is remarkable that 89 percent of the newborns in our cohort presented jaundice; however, they were not followed up at INPer. Generally, such cases are referred to pediatric hospital care centers.

Prevention strategies for viral hepatitis have been shown to be cost-effective. In Mexico, the vaccination strategy for HBV has been applied for more than two decades, showing successful results in the form of a very low prevalence of infection. Low numbers of pregnant women are affected, diminishing the probability of MTCT. At the same time, this strategy impacts the prevalence of HDV. In the case of HCV eradication, the recently implemented strategy to cure HCV with DAA treatment is expected to be successful and eventually impact on the prevalence of chronic liver disease, which is still among the most common causes of mortality in our country. Adequate follow-up should be given to these patients, as some liver damage, including HCC, can progress even after the HCV infection has been cured. Nevertheless, simple strategies should not be overlooked: education and hygiene programs, including information on hand washing, food and water safety, risk behavior awareness, safe sex, needle sharing, and maternal and donor blood testing, are also effective as a form of prevention in a wide part of the population. For the prevention of HAV and HEV, and to reduce their incidence, it will be important to establish vaccination strategies. In addition, increased detection of the different serotypes of HV, especially for HEV infections, must be ensured. Finally, patients who are positive for a hepatitis virus need to be effectively treated and tracked due to their public health impact.

## Figures and Tables

**Table 1 pathogens-13-00651-t001:** General characteristics of pregnant Mexican women with viral hepatitis.

Characteristic/Type HV	HAV	HBV	HCV	Total
**Number**	11 (28.9%)	11 (28.9%)	16 (42.1%)	38
**Maternal age**	24.91 (26)	24.45 (24)	34.31 (35)	29 ± 6.7
Advanced maternal age ≥ 35 years	0	3 (27.3%)	9 (56.25%)	12 (31.6%)
**Parity**				
First	2 (18.2%)	5 (45.5%)	5 (31.3%)	12 (31.6%)
Second	5 (45.5%)	2 (18.2%)	6 (37.5%)	13 (34.2%)
Third or more	4 (36.4%)	4 (36.4%)	5 (31.3%)	11 (29%)
**Previous miscarriages**	1 (9.1%)	2 (18.2%)	8 (50%)	13 (34.2%)
1	0	2 (18.2%)	6 (37.5%)	8(21.1%)
≥2	1 (9.1%)	0	2 (12.5%)	3 (7.9%)
**Hepatitis diagnosed**				
Before pregnancy	0	1 (9.1%)	5 (31.3%)	6 (15.8%)
By serological screening	11 (100%)	9 (81.8%)	11 (68.8%)	31 (81.6%)
By liver biopsy	0	1 (9.1%)	0	1 (2.6%)
**Weeks of gestation at diagnosis**	25 ± 8.03	20.36 ± 8.74	21.5 ± 12.64	23 ± 8.6
**Asymptomatic patients**	0	10 (90.9%)	13 (81.3%)	23 (60.5%)
**Symptomatic patients**	11 (100%)	1 (9.1%)	3 (18.7%)	15 (39.5%)
Jaundice	9 (81.8%)	1 (9.1%)	1 (6.2%)	11
Edema	2 (18.2%)	1 (9.1%)	3 (18.7%)	6
Abdominal pain	4 (36.4%)	0	0	4
Choluria/acholia	6 (54.5%)	0	0	6
Vomit/nausea	5 (45.5%)	0	0	5
Fever	3 (27.3%)	0	0	3
**Laboratory parameters**				
AST > 3 0 μ/L	3 (27.3%)	2 (18.2%)	3 (18.7%)	8
>300 μ/L	4 (36.4%)	0	0	4
ALT > 30 μ/L	1 (9.1%)	4 (36.4%)	4 (25%)	9
>300 μ/L	6 (54.5%)	0	0	6
LDH > 400 μ/L	2 (18.2%)	1 (9.1%)	4 (25%)	7
TBIL > 2 mg/dL	7 (63.6%)	2 (18.2%)	0	9
DBIL > 0.5 mg/dL	7 (63.6%)	3 (27.3%)	2 (12.5%)	12
**DBIL**	9.2 ± 23.24	0.53 ± 0.24	0.33 ± 0.29	*p* < 0.05

Data are expressed as mean + standard deviation or n and percentage (%); maternal age is represented by mean (median). All the patients were evaluated at the Gynecology and Obstetrics Department of INPer. N = 38. ALT: alanine aminotransferase; AST: aspartic aminotransferase; DBIL: direct bilirubin; TBIL: total bilirubin; LDH: lactic dehydrogenase.

**Table 2 pathogens-13-00651-t002:** Most frequent comorbidities and complications in pregnant Mexican women with viral hepatitis.

Characteristic/Virus Type	HAV	HBV	HCV	Total
**Number**	11 (28.9%)	11 (28.9%)	16 (42.1%)	38
**Comorbidities**				
Obesity	1 (9.1%)	1 (9.1%)	3 (18.8%)	5 (13.2%)
Systemic hypertension	1 (9.1%)	0	1 (6.25%)	2 (5.3%)
Hypothyroidism	0	1 (9.1%)	1 (6.25%)	2 (5.3%)
Cholelithiasis	0	1 (9.1%)	1 (6.25%)	2
Toxicomania	0	1 (9.1%)	1 (6.25%)	2
DM2	0	0	1 (6.25%)	1 (2.6%)
HIV	0	1 (9.1%)	0	1 (2.6%)
**Pregnancy related**				
ICP	3 (27.3%)	0	2 (12.5%)	5 (13.2%)
Preeclampsia	0	4 (36.4%)	1 (6.25%)	5 (13.2%)
**Without comorbidities**	2 (18.2%)	3 (27.3%)	3 (18.8%)	8 (21.1%)
**Complications**				
Gestational diabetes	2 (18.2%)	5 (45.5%)	2 (12.5%)	9 (23.7%)
Thrombocytopenia	1 (9.1%)	0	5 (31.3%)	6 (15.8%
Anemia	0	1 (9.1%)	1 (6.25%)	2 (5.3%)
Glomerulopathy	0	1 (9.1%)	0	1 (2.6%)
PRM	2 (18.2%)	0	1 (6.25%)	3 (7.9%)
Postpartum hemorrhage	3 (27.3%)	1 (9.1%)	3 (18.8%)	7 (18.4)
**Without maternal complications**	4 (36.4%)	5 (45.5%)	5 (31.3%)	14 (36.8%)

DM2: diabetes mellitus type 2; ICP: intrahepatic cholestasis of pregnancy; PRM: premature rupture of membranes.

**Table 3 pathogens-13-00651-t003:** Pregnancy and neonatal outcomes in Mexican women with viral hepatitis.

Characteristic/Virus Type	HAV	HBV	HCV	Total
**Number**	11	11	16	38
**Pregnancy outcomes**				
Pregnancy length (weeks)	36.6 (31–39)	36.7 (27–40)	37.2 (29–40.3)	36 ± 4.4
Singleton	10	10	16	36/38 (94.7%)
Twin	1	1	0	2/38 (5.3%)
Term births (≥37 weeks)	5	8	11	24/35 (63.2%)
Preterm births (<37 weeks)	4	3	4	11/35 (28.9%)
Unspecified pregnancy length	2	0	0	2/38 (5.2%)
Miscarriage	0	0	1	1/38 (2.6%)
Vaginal delivery	3	3	3	9/36 (23.7)
Cesarean section	7	8	12	27/36 (74.7%)
Not specified	1	0	0	1
**Neonatal outcomes/Number**	12	12	15	39 babies
Preterm babies (<37 weeks)	5/10 (50%)	4 (33.3%)	4 (26.7%)	14/37 (37.83%)
Premature babies (≤33 weeks)	1/10 (10%)	1 (8.3%)	2 (14.3%)	4/37 (10.8%)
Small for gestational age	4/12 (33.33%)	3 (25%)	1 (6.7%)	8/39 (20.5%)
Large for gestational age	0	0	2 (13.3%)	2/39 (5.1%)
Jaundice	8/9 (88.9%)	6/10 (60%)	8/12 (66.7%)	22/32 (68.8%)
Without jaundice	1	4	4	9/32 (28.1%)
Nondeterminate	2	2	3	7/39
NICU admission	1	0	1	2/39 (5.1%)
Transition unit care	0	3	5	8/39 (20.5%)

Note: Pregnancy length shows the mean (range); in the Total column, data are expressed as mean ± standard deviation. NICU: neonatal intensive care unit.

**Table 4 pathogens-13-00651-t004:** Placental histopathological lesions from pregnancies with viral hepatitis.

Placental Characteristic/Hepatitis Type	HAV	HBV	HCV	Total
**Number**	11	10	14	35
**Normal**	0	0	1 (7.1%)	1 (2.9%)
**Hypertrophy**	1 (9.1%)	0	4 (28.6%)	5 (14.3%)
**Histopathology**				
**Maternal Vascular Malperfusion**	3 (27.3%)	3 (30%)	1 (7.1%)	7 (20%)
Hypotrophy	2 (18.2%)	3 (30%)	2 (14.3%)	7 (20%)
Infarcts	2 (18.2%)	1 (10%)	0	3 (8.6%)
Retroplacental hemorrhage	5 (45.5%)	1 (10%)	1 (7.1%)	7 (20%)
AVM without ISK	4 (36.4%)	0	0	4 (11.4%)
AVM + ISK	0	1 (10%)	1 (7.1%)	2 (5.7%)
ISK without AVM	3 (27.3%)	2 (20%)	1 (7.1%)	6 (17.1%)
**Fetal Vascular Malperfusion**			1 (7.1%)	
Intervillous fibrinoid	6 (54.5%)	3 (30%)	2 (14.3%)	11 (31.4%)
Decidual arteriopathy	0	0	3	3 (8.6%)
Stem vessel obliteration	6 (54.5%)	1 (10%)	2 (14.3%)	9 (25.7%)
**MVM and FVM**			1 (7.1%)	1 (2.9%)
**Delayed Villous Maturation**	4 (36.4%)	3 (30%)	5 (35.7%)	12 (34.3%)
**Acute Intrauterine Infection**				
Chorioamnionitis	1 (9.1%)	1 (10%)	1 (7.1%)	3 (8.6%)
**Microcalcifications**	2 (18.2%)	2 (20%)	3 (21.4%)	7 (20%)
**Chorangiosis**	5 (45.5%)	3 (30%)	7 (50%)	15 (42.9%)

Note: We obtained histopathological data from 35 placentas, including the ones from the twin pregnancies and the miscarriage. Only one placenta was reported as normal, AVM: accelerated villous maturation; DVM: Delayed Villous Maturation; FVM: fetal vascular malperfusion; ISK: increased syncytial knots; MVM: maternal vascular malperfusion.

## Data Availability

Data available upon direct request.

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
