# Peer review of "Viral Hepatitis in Pregnant Mexican Women: Its Impact in Mother–Child Binomial Health and the Strategies for Its Eradication"

_pathogens, 2024, doi:10.3390/pathogens13080651_

Round 1

Reviewer 1 Report

Comments and Suggestions for Authors

Specific Comments on the Manuscript

Line 88: Are you calculating incidence or prevalence? Please use the appropriate epidemiological term.

Statistical analysis section is entirely missing.

Ethical approval is entirely missing.

Anonymization of the subjects is entirely missing.

Criteria such as STROBE for conducting the study is entirely missing as well as all the things that you should mention according to these criteria.

Limitation section is entirely missing

Author Response

Answers to reviewer 1

According to your suggestions, minor changes have been done.

Major suggestions have been resolved, including changes in the methodology sections, in particular, we included now a statistical analysis section.

Ethical aspects were already re written.

According to your comments about STROBE criteria, we have already used the check list for cohort studies.

Finally, we included in the text a paragraph specifying weakness and limitation of the study.

All changes of the manuscript were highlighted.

Reviewer 2 Report

Comments and Suggestions for Authors

This manuscript describes epidemiological data obtained from electronic medical records on viral hepatitis in Mexican pregnant women. This topic is important and the manuscript is well written. However, the study has several limitations.  

Comments :

1.       The design of this study must be clarified. There were 17 206 pregnancies from 1 January 2018 to December 2023. Were all pregnant women (with or without clinical symptoms) tested for HBV and HCV ? Were all pregnant women with clinical symptoms (jaundice) tested for HAV and HEV ? Were pregnant women tested for all hepatitis viruses when there was an ALT elevation ?

2.       HEV is an important pathogen during pregnancy. Unfortunately, the data presented in this paper are very limited. In addition, the clinical impact of HEV infection is strongly related to HEV genotype (more severe disease for HEV genotypes 1 and 2 compared to HEV genotypes 3 and 4). According to current litterature, HEV genotypes 1 and 3 are circulating in Mexico and HEV-2 circulated in the past. Recent data on IgM anti-HEV indicating recent infection could be useful. Indeed, IgG seroprevalence is strongly influenced by the type of test (Hartl Viruses 2016). Therefore, the prevalence of 0.22 % for viral hepatitis in Mexican pregnant women could be underestimated because hepatitis E is the most frequent viral hepatitis in many countries.

3.       Minor points :

-          Introduction, line 70 : « Infection with HEV genotype 1/2… » rather than « Infection with HEV… »

-          Discussion, line 275 : distinct placental alterations and level of replication have been described in experimental models for HEV genotypes 1 and 3 (Gouilly 2018).

Author Response

Answers to reviewer 2

According to your comment 1, the design of our study was clarified, including clinical symptoms and biochemical determinations.

Your suggestions about HEV genotypes, we have mentioned in the introduction and discussion sections, included the suggested and new references.

Minor point that suggested change in line 17 has been done.

All changes of the manuscript were highlighted.

Round 2

Reviewer 1 Report

Comments and Suggestions for Authors

All relevant amendments have been performed.

Reviewer 2 Report

Comments and Suggestions for Authors

The manuscript has been improved. The answers are globally satisfactory.